# An NF-κB-microRNA regulatory network tunes macrophage inflammatory responses

Mati Mann[1], Arnav Mehta[1,2], Jimmy L. Zhao[3,4], Kevin Lee[1], Georgi K. Marinov [1], Yvette Garcia-Flores[1], Li-Fan Lu[5,6,7], Alexander Y. Rudensky[8] & David Baltimore[1]

The innate inflammatory response must be tightly regulated to ensure effective immune protection. NF-κB is a key mediator of the inflammatory response, and its dysregulation has been associated with immune-related malignancies. Here, we describe a miRNA-based regulatory network that enables precise NF-κB activity in mouse macrophages. Elevated miR-155 expression potentiates NF-κB activity in miR-146a-deficient mice, leading to both an overactive acute inflammatory response and chronic inflammation. Enforced miR-155 expression overrides miR-146a-mediated repression of NF-κB activation, thus emphasizing the dominant function of miR-155 in promoting inflammation. Moreover, miR-155-deficient macrophages exhibit a suboptimal inflammatory response when exposed to low levels of inflammatory stimuli. Importantly, we demonstrate a temporal asymmetry between miR-155 and miR-146a expression during macrophage activation, which creates a combined positive and negative feedback network controlling NF-κB activity. This miRNA-based regulatory network enables a robust yet time-limited inflammatory response essential for functional immunity.

[1] Division of Biology and Biological Engineering, California Institute of Technology, Pasadena, CA 91125, USA. [2] David Geffen School of Medicine, University of California, Los Angeles, CA 90095, USA. [3] Department of Medicine, New York Presbyterian Hospital, Weill Cornell Medical College, 525 E 68th Street, New York, NY 10065, USA. [4] Division of Hematology Oncology, Memorial Sloan Kettering Cancer Center, 1275 York Avenue, New York, NY 10065, USA. [5] Division of Biological Sciences, University of California, La Jolla, San Diego, CA 92093, USA. [6] Moores Cancer Center, University of California, La Jolla, San Diego, CA 92093, USA. [7] Center for Microbiome Innovation, University of California, La Jolla, San Diego, CA 92093, USA. [8] Howard Hughes Medical Institute and Immunology Program, Ludwig Center at Memorial Sloan-Kettering Cancer Center, Memorial Sloan-Kettering Cancer Center, New York, NY 10065, USA. Correspondence and requests for materials should be addressed to M.M. (email: mati@caltech.edu) or to D.B. (email: baltimo@caltech.edu)

Inflammation is initiated by innate immune cells in response to external stimuli such as pathogen-associated molecular patterns or host-derived damage-associated molecular patterns. These cells initiate signalling cascades that activate key transcription factors and regulators such as NF-κB, AP1 and MAPKs, all of which regulate inflammation-specific genes[1,2]. During an acute inflammatory response, it is crucial that immune cells respond quickly and efficiently to overcome the early expansion of pathogens. It is also crucial for the inflammatory response to be tightly regulated to avoid tissue damage and septic shock. Dysregulated innate inflammatory responses can also gradually develop into chronic, low-grade inflammation by constant production of cytokines and reactive oxygen species at low levels[3]. With time, chronic inflammation may cause dysregulated adaptive immunity that precedes to autoimmunity, heart disease and cancer (reviewed in refs. [3,4]). To prevent this, many inflammatory cascades have built-in feedback mechanisms to both positively and negatively regulate inflammatory signalling, thus allowing a pulsatile response with rapid induction of inflammation followed by return of the response to pre-stimulation levels. One of the best-described examples of such feedback regulation is in NF-κB signalling[5]. Both positive and negative feedback mechanisms have been described that ensure appropriate NF-κB activity during an inflammatory response[6–10].

It had become evident that microRNAs (miRNAs) are pivotal regulators of many biological processes, including inflammation. We and others have shown that several miRNAs function in both positive and negative regulation of the inflammatory response, participating in various regulatory network motifs (reviewed in ref. [11]). Two particular miRNAs, miR-155 and miR-146a, have been extensively characterized[12–15]. Both miR-146a and miR-155 are transcriptionally regulated by NF-κB and induced in macrophages following Toll-like receptor (TLR) activation[12,16]. miR-146a functions as an anti-inflammatory regulator in various immune cell types by repressing NF-κB and AP1 signalling[17], and has been shown to be involved in the regulation of the acute inflammatory response and endotoxin tolerance[18]. We have shown that miR-146a-deficient (*miR-146a*$^{-/-}$) mice serve as a genetic model for low grade, chronic inflammation because they have supraphysiological levels of serum autoantibodies and interleukin (IL)-6 with age, and develop myeloproliferative disorders and cancers[13,17]. miR-155 expression, on the other hand, has been shown to be essential for T-cell, B-cell and myeloid cell development and function[14,15,19–21]. miR-155 was originally discovered as its primary transcript, BIC, which was deregulated in B-cell malignancies and leukaemia[22]. Similarly to what is observed in *miR-146a*$^{-/-}$ mice, enforced expression of miR-155 in the bone marrow compartment causes myeloproliferation and cancers[23]. It was shown that miR-155 has an epistatic function over miR-146a during T-cell antitumour responses and T-follicular helper cell development[24,25]. However, the interrelationship of miR-155 and miR-146a in the regulation of inflammatory responses remains to be investigated.

Here, we aim to characterize the genetic and functional interaction between miR-146a, miR-155 and NF-κB activity at homeostasis, and during acute inflammatory stimuli in macrophages. We show that miR-155 and miR-146a coordinately regulate the macrophage inflammatory response by forming a combined negative and positive regulatory loop that alters NF-κB activity. This network architecture enables a defined inflammatory response, beginning with robust induction of NF-κB signalling and a precisely timed shutdown dynamics. We also show that dysregulated levels of miR-146a and miR-155 can cause a suboptimal immune response following low doses of inflammatory stimuli, or to the development of chronic inflammation, which in turn may lead to the induction of myeloproliferation and extramedullary haematopoiesis.

## Results

### miR-155 is required for *miR-146a*$^{-/-}$ pathology in aged mice.
To determine the miR-155 and miR-146a genetic hierarchy at steady state, we followed wild-type (WT), *miR-155*$^{-/-}$, *miR-146a*$^{-/-}$, and *miR-155*$^{-/-}$ *miR-146a*$^{-/-}$ (double knockout (DKO)) mice for up to 12 months. As we and others previously reported, miR-146a deficiency in aged mice leads to low-grade chronic inflammation manifested by increased levels of serum IL-6, elevated splenic TNF, IL-1β and IL-6 messenger RNA (mRNA) expression, myeloid cell expansion, extramedullary haematopoiesis, and enlarged spleens[13,17] (Fig. 1a–f). These phenotypes do not appear in young mice, and gradually progress with age beginning with myeloid skewing at 4–5 months of age (Supplementary Fig. 1A–D). Aged *miR-155*$^{-/-}$ mice were comparable to WT mice with a slight but significant reduction in spleen CD11b$^+$F4/80$^+$ macrophages (Fig. 1e). Notably, aged DKO mice did not present any of the phenotypes found in *miR-146a*$^{-/-}$ mice, and phenocopied WT and *miR-155*$^{-/-}$ mice, thus implying that miR-155 expression is required for miR-146a pathology (Fig. 1a–f). Since the first detectable haematopoietic phenotype in aging *miR-146a*$^{-/-}$ mice is myeloid lineage cell excess, we examined if knocking out miR-146a only in the myeloid lineage is sufficient to recapitulate the total KO phenotype. For that, we established myeloid lineage-specific miR-146a-deficient mice by crossing *miR-146a* floxed (fl) mice with LyzM-Cre mice. *LyzM-Cre miR-146a*$^{fl/fl}$ mice breed normally and present a normal immune cell panel comparable to WT mice until 6 months of age (Supplementary Fig. 1A–D). Similar to *miR-146a*$^{-/-}$ mice, deleting miR-146a just in the myeloid lineage leads to a myeloid bias, an enlarged spleen and mild extramedullary haematopoiesis with age (Fig. 1c–f). These results indicate that miR-146a deficiency only in the myeloid lineage is sufficient for the development of systemic chronic inflammation with age, albeit to a lesser extent than the complete knockout. We therefore focused on the myeloid lineage, and tried to decipher the interplay between miR-155 and miR-146a in the regulation of acute and chronic inflammation.

### miR-155 is required for *miR-146a*$^{-/-}$ acute inflammatory phenotypes.
We next characterized the contribution of miR-155 and miR-146a to the acute innate inflammatory response. In an effort to understand the initial contribution of miR-146a and miR-155 to immune function, we used 8–10-week-old mice (young mice), before the characteristic myeloproliferation or the chronic inflammation phenotype is manifest in *miR-146a*$^{-/-}$ mice. We infected cohorts of WT, *miR-155*$^{-/-}$, *miR-146a*$^{-/-}$ and DKO mice with a low, sublethal dose of an attenuated strain of *Listeria monocytogenes* (strain 10,403 serotype 1, 10E5 c.f.u.). At 72 h after infection, the time of the maximal innate response and before the onset of the adaptive response[26], we collected the mice and quantified the innate immune response by measuring IL-6 serum levels, CFUs of bacteria in the spleen and liver, and haematopoietic cell populations using flow cytometry (fluorescence-activated cell sorting (FACS)). While the numbers and percentage of peripheral blood macrophages did not change significantly between strains (Supplementary Fig. 2A), *miR-146a*$^{-/-}$ mice had distinctly lower bacterial loads in both the spleen and liver, higher serum IL-6 levels and lost more weight compared to WT mice (Fig. 2a–c; Supplementary Fig. 2B). WT, *miR-155*$^{-/-}$ and DKO mice displayed similar bacterial loads and weight indices. IL-6 serum levels in *miR-155*$^{-/-}$ mice were slightly but significantly lower than in WT and DKO mice (Fig. 2c). Similar results were also obtained using bone marrow-derived macrophages (BMMs)

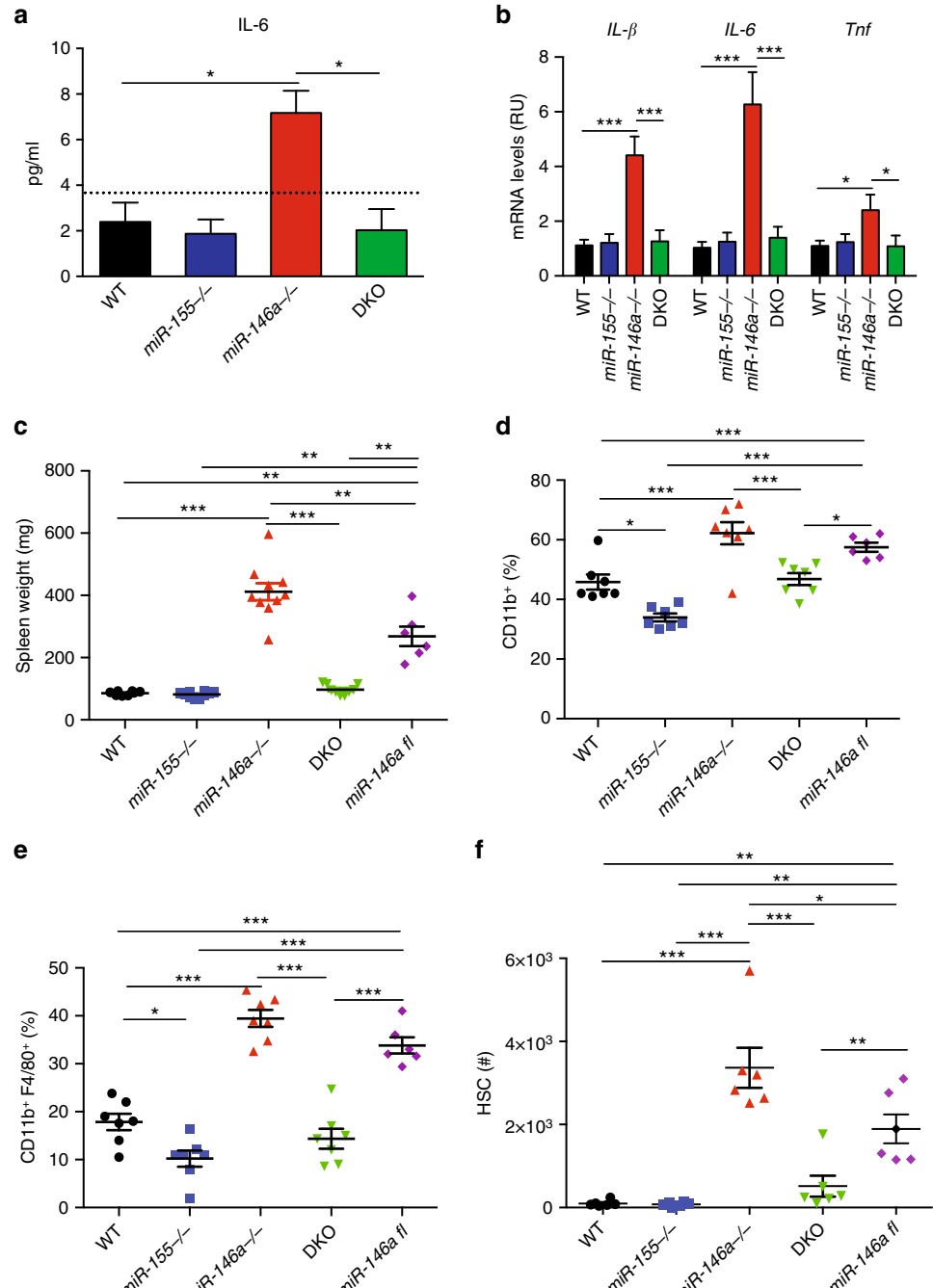

**Fig. 1** miR-155 is required for myeloproliferation and extramedullary haematopoiesis in aged *miR-146a⁻/⁻* mice. **a**, **b** Twelve-month-old WT, *miR-155⁻/⁻*, *miR-146a⁻/⁻* and double knock out (DKO) mice were analysed for serum IL-6 levels (**a**), spleen mRNA levels of *IL-1b*, *IL-6* and *TNF* (**b**). **c**–**f** Twelve-month-old WT, *miR-155⁻/⁻*, *miR-146a⁻/⁻*, DKO and *LyzM-Cre miR-146a^{fl/fl}* mice were analysed for spleen weight (**c**), relative macrophage percentage in peripheral blood (**d**) and spleen (**e**), as well as extramedullary haematopoiesis by HSC quantification in spleen (**f**). N = 10 (**a**, **b**); N = 7 (**c**, **d**) per group, from at least two independent experiments and are represented as mean ± SEM. *p < 0.05, **p < 0.01 and ***p < 0.001 using one-way ANOVA

infected with *Salmonella* Typhimurium. Infection with live Gram-negative bacteria had led to an elevated inflammatory response in *miR-146a⁻/⁻* BMMs, manifested by elevated CD80 and MHC-II cell surface expression compared to WT, *miR-155⁻/⁻* and DKO BMMs, which displayed similar response. No significant difference in cell death or proliferation was observed between strains (Supplementary Fig. 2C–F).

Endotoxin tolerance is a well-defined example of an intracellular mechanism for inflammation resolution, where cells become refractory to subsequent endotoxin challenge after an initial challenge. Because *miR-146a⁻/⁻* mice demonstrated an elevated acute inflammatory response to *Listeria* challenge and miR-146a has been shown to participate in the regulation of endotoxin tolerance[18], we examined the interplay of miR-146a and miR-155 in this phenomenon. We assayed in vivo endotoxin tolerance by three serial injections of 1 mg/kg lipopolysaccharide (LPS) every 24 h IP and monitored peripheral blood CD11b macrophage activation as well as serum IL-6 levels. All strains demonstrated endotoxin tolerance after the second and third injections with no significant difference in cell proliferation, CD11b frequency or

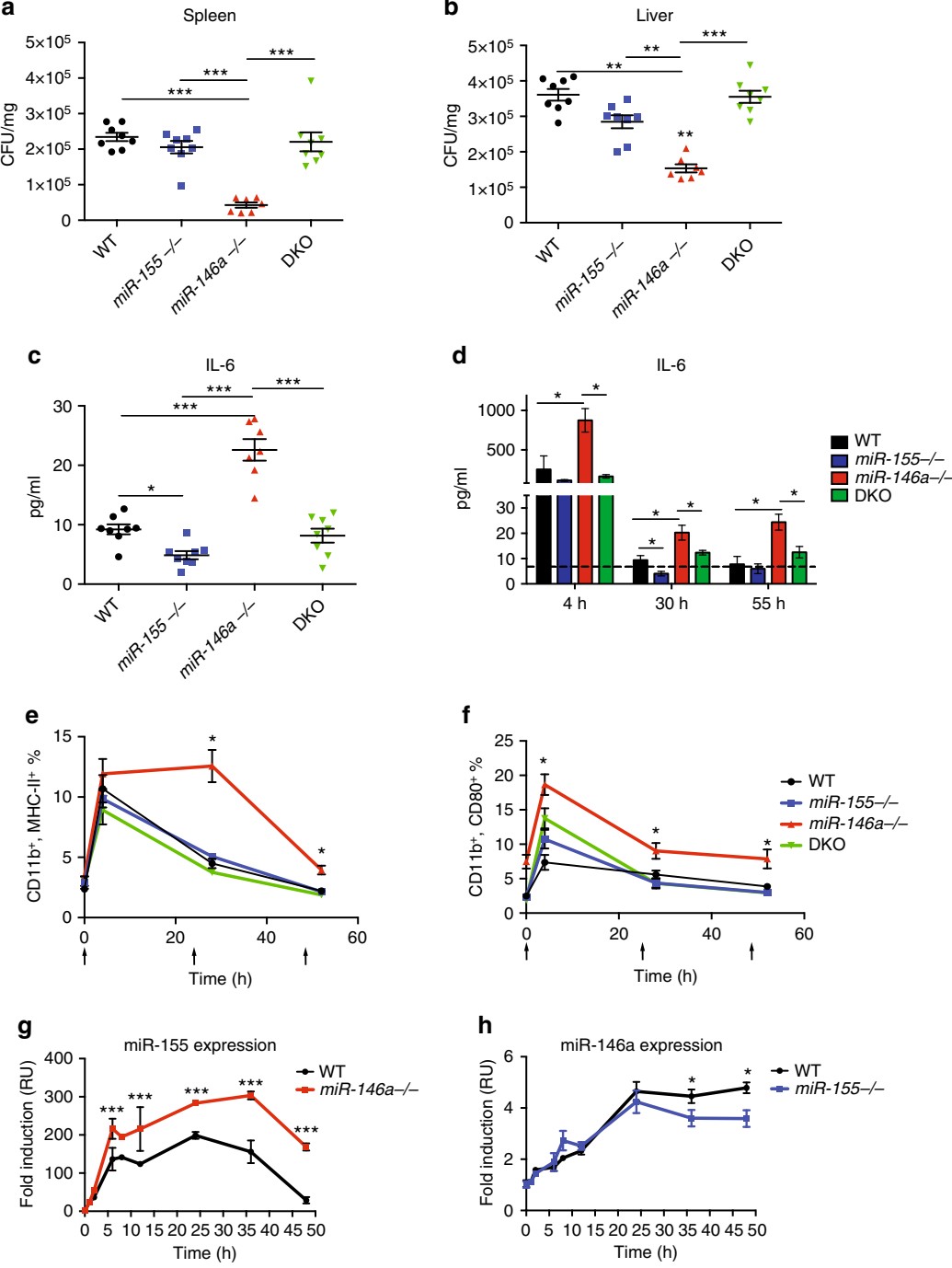

**Fig. 2** miR-155 expression is required for the elevated acute inflammatory response in miR-146a deficient mice. **a–c** Eight to ten-week-old WT, *miR-155*[−/−], *miR-146a*[−/−] and double knock out (DKO) mice were infected with *Listeria* monocytogenes. Seventy-two hours after infection, mice were analysed for colony formation units (CFUs) in spleen (**a**) and liver (**b**), as well as serum IL-6 levels (**c**). Eight to ten-week-old WT, *miR-155*[−/−], *miR-146a*[−/−] and DKO mice received 1 mg/kg LPS every 24 h for 3 days via IP injections. Serum IL-6 levels (**d**) as well as peripheral blood macrophage activation was quantified 4 h after each injection by CD11b[+] MHC-II high (**e**) and CD11b[+] CD80[+] percentage (**f**). Arrows indicate time of LPS injections. **g–h** Time course expression profile of miR-155 of WT and *miR-146a*[−/−] (**g**) and miR-146a of WT and *miR-155*[−/−] BMMs (**h**) after LPS stimulation (100 ng/mL) were quantified by qPCR. N > 7 (**a–d**), or N > 9 (**e–h**) per group from two independent experiments and are represented as mean ± SEM. *$p < 0.05$, **$p < 0.01$ and ***$p < 0.001$ using one-way ANOVA (**a–d**) or two-way ANOVA (**d–h**)

cell death (Supplementary Fig. 3A–C). However, *miR-146a*[−/−] mice presented a stronger and prolonged acute macrophage response as measured by the activation markers CD80[+] and MHC-II, and by serum IL-6 levels in comparison to WT, *miR-155*[−/−] and DKO mice. As with *Listeria* infection and the chronic inflammatory phenotype, ablating miR-155 completely rescued

the *miR-146a*[−/−]-enhanced inflammatory response, leading to WT levels of response in all measurements (Fig. 2d–f and Supplementary Fig. 3A–C).

These results were also replicated in BMMs stimulated with LPS for up to 48 h in vitro, showing elevated and prolonged *IL-6* and *IL-1β* mRNA expression, and elevated CD80 and MHC-II

cell surface expression in *miR-146a*$^{-/-}$ BMMs compared to WT, *miR-155*$^{-/-}$, and DKO mice. No significant differences in cell death or proliferation were observed between strains (Supplementary Fig. 3D–I).

These results indicate that miR-146a functions as a negative regulator of miR-155 expression and the innate inflammatory response, and that miR-155 expression is essential for the inflammatory phenotype observed in *miR-146a*$^{-/-}$ mice, which can lead to enhanced myeloid proliferation at later stages. This interdependency between miR-146a and miR-155 implies that they co-participate in the regulation of the inflammatory response and that miR-155 plays a dominant positive role in this regulation.

**miR-155 overexpression enhances NF-κB activity.** Both miR-146a and miR-155 are regulated through the activity of the NF-κB transcription factor[12], and since miR-146a attenuates NF-κB activity, we wished to determine whether miR-155 expression is affected by miR-146a repression. For that, we quantified miR-155 expression dynamics in BMMs of WT and *miR-146a*$^{-/-}$ mice following a single dose of LPS. As shown in Fig. 2g, in WT BMMs, miR-155 reaches a peak level by 12–24 h and returns to near basal levels by 48 h. In *miR-146a*$^{-/-}$ BMMs, miR-155 rises more rapidly and declines more slowly; miR-155 stays at what would be peak WT levels even after 48 h, a time when WT macrophages have returned to baseline. These results indicate that miR-146a is a major controller of miR-155 expression during the inflammatory response, and that elevated and prolonged expression of miR-155 correlates with an elevated inflammatory response and chronic macrophage activation. We next quantified miR-146a expression dynamics in WT and *miR-155*$^{-/-}$ BMMs. As shown in Fig. 2h, in WT BMMs, miR-146a levels rise slower, starting at 8 h post stimulation, peak at 24 h and maintains peak levels until 48 h. In *miR-155*$^{-/-}$ BMMs, miR-146a expression dynamics followed that of WT BMMs with a slight attenuation starting 36 h after stimulation. These results imply that miR-146a functions at later stages following inflammatory stimulation, and that miR-155 expression may contribute to maintaining prolonged, high levels of miR-146a.

To further elucidate the genetic hierarchy between miR-155 and miR-146a in the regulation of NF-κB and the inflammatory response, and to understand whether miR-155 expression is sufficient to drive the inflammatory response observed in miR-146a deficient mice, we examined the effects of enforced expression of miR-155, miR-146a, or both miR-155 and miR-146a (dmiR) on myeloid proliferation and the inflammatory response. Lethally irradiated mice were reconstituted with WT bone marrow cells transduced with control (MG), miR-155, miR-146a or dmiR expressing vectors. Similar to miR-146a deficiency, we have shown that enforced miR-155 expression induces myeloproliferative disorders, as well as extramedullary haematopoiesis[23]. As expected, mice reconstituted with miR-155-expressing bone marrow displayed CD11b$^{+}$ myeloproliferation and enlarged spleens by 4 months post reconstitution (Fig. 3a, b). miR-146a overexpression did not lead to altered spleen size or mature immune cell composition in the spleen or peripheral blood. Interestingly, overexpression of both miR-146a and miR-155 led to enlarged spleens and myeloproliferation similar to miR-155 expressing mice (Fig. 3a, b). These results indicate that miR-155 expression plays a dominant role over miR-146a at steady state, leading to a myeloid bias.

We next determined whether miR-155 overexpression also leads to a prolonged inflammatory response and NF-κB activation as in *miR-146a*$^{-/-}$ mice. We used bone marrow from an NF-κB reporter mouse strain, which expresses GFP when

NF-κB-mediated transcription is activated. This mouse model enables monitoring of the dynamics of NF-κB activity in vivo (NF-κB-GFP, ref. [27]). NF-κB-GFP donor bone marrow cells were transduced with virus expressing miR-146a, miR-155, dmiR or control, and reconstituted into lethally irradiated WT mice. Peripheral blood from mice 3 months after reconstitution showed elevated NF-κB activity in CD11b$^{+}$ macrophages expressing miR-155 and dmiR, and slightly lower NF-κB activity in miR-146a expressing mice compared to WT (Fig. 3c). We next assayed the innate acute inflammatory response and endotoxin tolerance by three serial injections of LPS every 24 h and monitored peripheral blood CD11b macrophage activation as well as serum IL-6 levels. miR-146a overexpression led to attenuated NF-κB activity in CD11b$^{+}$ macrophages and serum IL-6 levels, while miR-155 overexpression led to increased and longer-lasting NF-κB activity and IL-6 serum levels compared to control mice. Similarly, overexpression of both miR-146a and miR-155 resulted in increased NF-κB activity and IL-6 serum levels, comparable to miR-155 overexpression (Fig. 3d, e).

Together, we show that miR-155 functions as a positive regulator, while miR-146a functions as a negative regulator of NF-κB activity and the inflammatory response. Our results indicate that miR-155 expression levels are regulated by miR-146a, and that elevated miR-155 levels can overcome miR-146a-mediated repression of NF-κB activity, suggesting that miR-155 acts downstream of miR-146a in the NF-κB signalling cascade.

**SHIP1 and SOCS1 repression by miR-155 regulates inflammation.** As negative regulators with a short mRNA-binding sequence, miRNAs have the potential to regulate numerous genes. We next set out to decipher miR-155 and miR-146a's role in the regulation of the macrophage inflammatory response, and to characterize their targets in this cell type. For that, we utilized RNA-sequencing on samples derived from BMMs of WT, *miR-146a*$^{-/-}$, *miR-155*$^{-/-}$ and dKO mice at steady state and 8 h after LPS stimulation. While we observed that both at steady state and after LPS stimulation, the expression profiles of all strains were broadly highly similar (Supplementary Fig. 4A, B), we also observed that *miR-146a*$^{-/-}$ BMMs were a clear outlier. Gene ontology analysis of genes upregulated in these cells revealed an enrichment in genes involved in cytokine-related activity, phagocytosis, cytokine production and the innate immune response when compared to WT, *miR-155*$^{-/-}$ and dKO BMMs (Fig. 4b, Supplementary Data 1). *miR-155*$^{-/-}$ macrophages were similar to WT macrophages, but still presented a differential gene response to LPS, mainly with genes related to cell motility and cell migration (Supplementary Data 1).

We next analysed the expression of all known miR-155 and miR-146a targets (based on TargetScan[28]). *Irak1* and *Traf6* were among the genes that were significantly upregulated in *miR-146a*$^{-/-}$ vs. WT BMMs (Supplementary Table 1). miR-146a is known to directly repress expression of *Traf6* and *Irak1*, two crucial adaptors for TLR-mediated NF-κB signalling[1]. Indeed, we found that both *Traf6* and *Irak1* mRNA levels are higher in *miR-146a*$^{-/-}$ and DKO mice compared to WT BMMs using qPCR (Fig. 4c). Interestingly, although *Traf6* and *Irak1* levels are high in DKO mice, these mice do not show NF-κB elevation or myeloid activation, emphasizing the critical role of miR-155 in producing these effects.

Analyzing miR-155-predicted targets, we found upregulation of several genes in *miR-155*$^{-/-}$ cells, including *Pu.1*, *Bach1*, *Ets1*, *Il21*, *Socs1* and *Ship1* (Supplementary Table 1). After verification of differential expression using qPCR, we focused on *Bach1*, *Ship1* and *Socs1* as the most robustly and significantly differentially expressed genes in our system (Fig. 4c). In macrophages, it has been

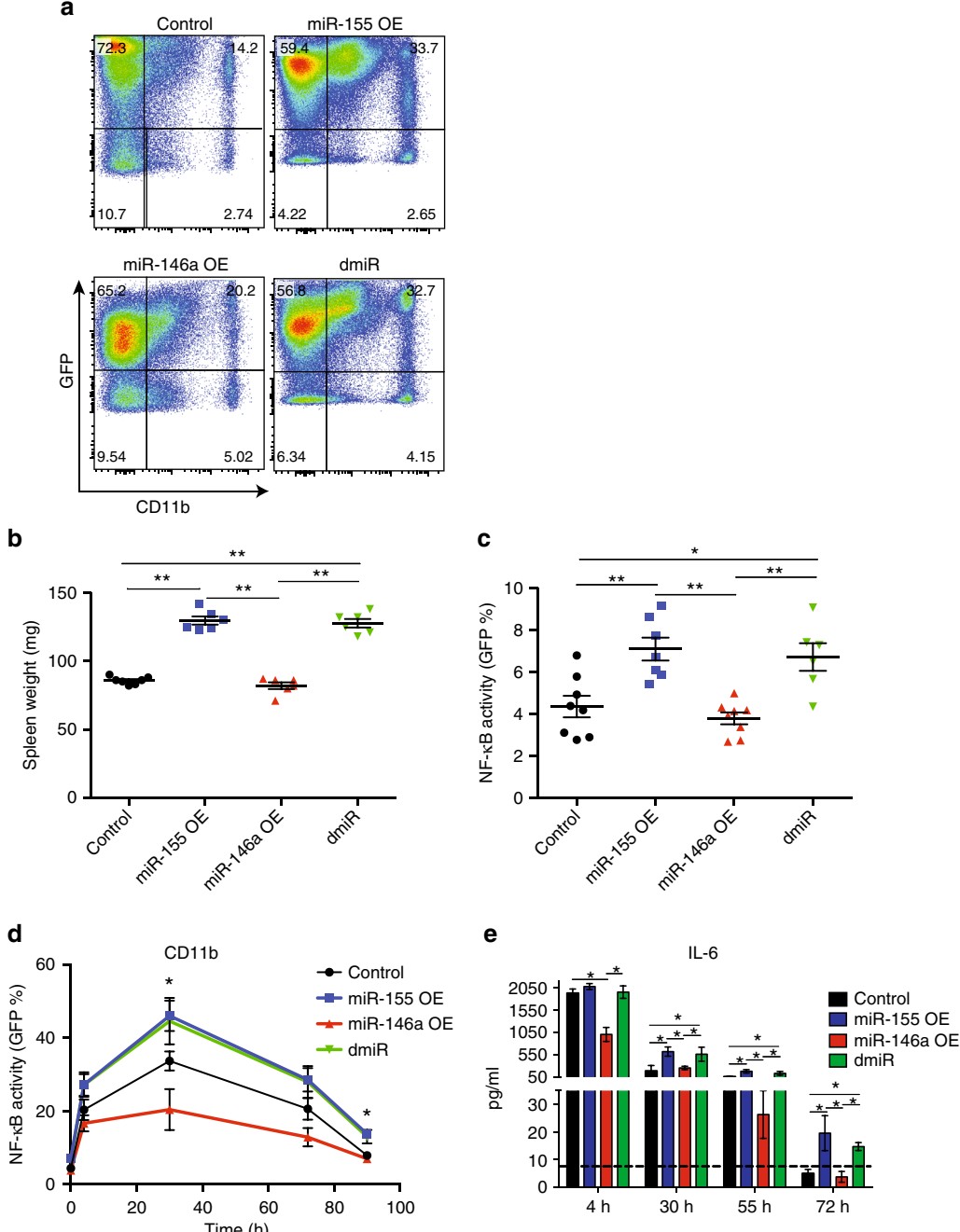

**Fig. 3** MiR-155 overexpression overrides miR-146a repression of NF-κB, leading to NF-κB activation and myeloproliferation. **a**, **b** Mice reconstituted with bone marrow transduced with GFP and control (MG), miR-155, miR-146a or both miR-155 and miR-146a (dmiR) were analysed 4 months after reconstitution for (**a**) CD11b[+] peripheral blood macrophages and (**b**) spleen weight. **c–e** Mice reconstituted with bone marrow from NF-κB reporter mice, transduced with control (MG), miR-155, miR-146a or dmiR were analysed 3 months after reconstitution. (**c**) Basal NF-κB activity of CD11b[+] peripheral blood macrophages. (**d**) Mice received 1 mg/kg LPS every 24 h for 3 days via IP injections. Peripheral blood macrophage NF-κB activation was quantified 4 h after each injection by GFP expression, as well as serum IL-6 levels (**e**). Dashed line represent ELISA detection sensitivity. N = 7 per group from three (**a–c**) or two (**d**, **e**) independent experiments and are represented as percentage (**a**) or mean ± SEM. *p < 0.05, **p < 0.01, using one-way ANOVA (**b**, **c**) or two-way ANOVA (**d–e**)

previously shown that miR-155 directly regulates the expression of *Ship1* and *Socs1*, two negative regulators of the macrophage inflammatory response. Both genes were reported to regulate NF-κB activity as well as endotoxin tolerance[29,30]. Indeed, both *Ship1* and *Socs1* mRNA and protein levels are higher in *miR-155*[−/−] and DKO BMMs after LPS stimulation (Fig. 4c, d). Interestingly, SHIP1 and SOCS1 protein levels were slightly but significantly lower in *miR-146a*[−/−] BMMs compared to WT, in line with the higher

expression of miR-155 in these cells (Supplementary Fig. 5A; Fig. 2g). To examine whether *Ship1* or *Socs1* play a role in miR-155 regulation of NF-κB activity in vivo, we knocked down either *Ship1* or *Socs1* in WT, *miR-155*[−/−], *miR-146a*[−/−] and DKO bone marrow cells using short hairpin RNAs (shRNAs). These cells were then used to reconstitute the immune system of lethally irradiated C57BL/6 WT mice. The knockdown levels of both *Ship1* and *Socs1* were about 50%, similar to the levels of repression achieved in WT

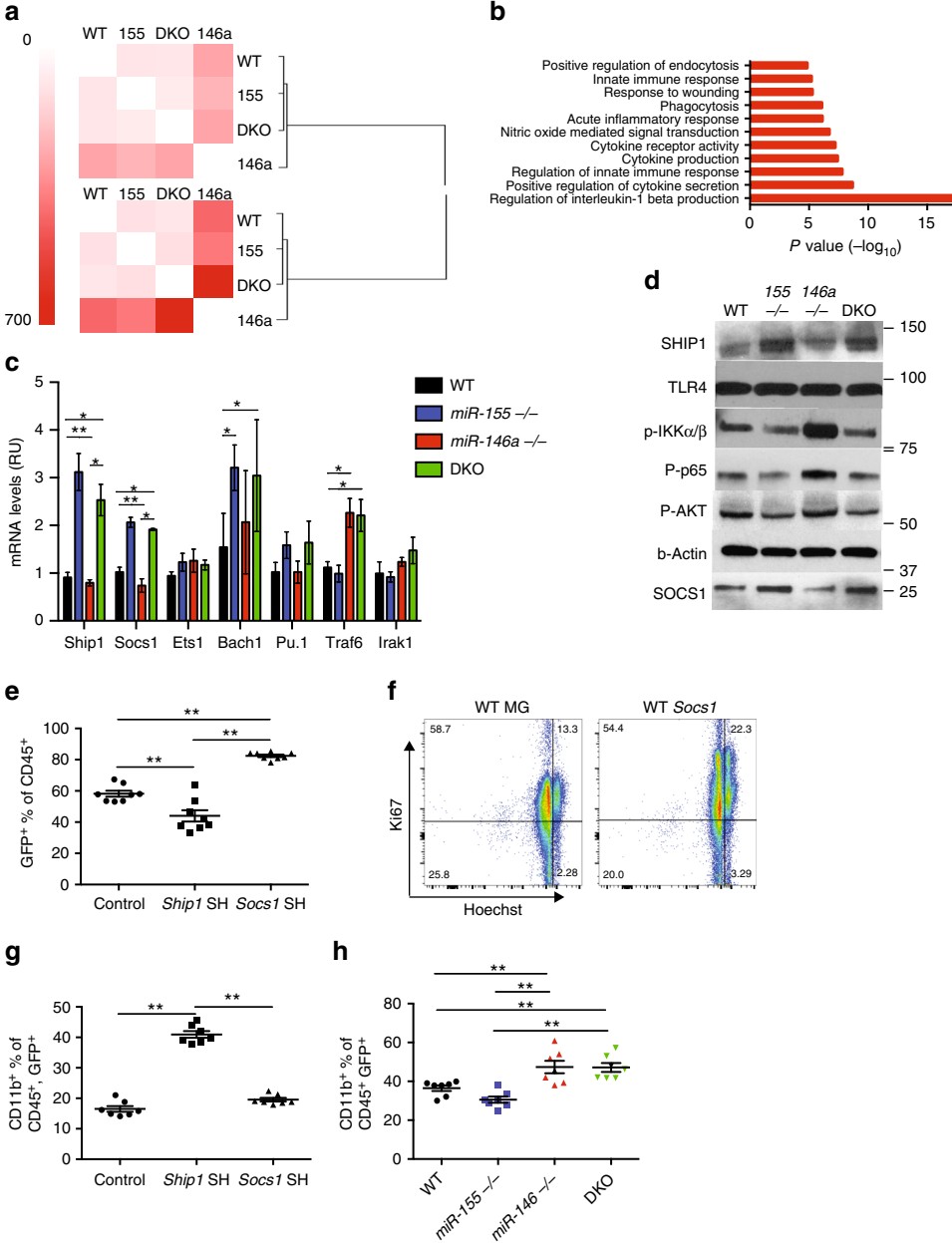

**Fig. 4** Molecular characterization of *miR-146a* and *miR-155* KO BMMs reveal SHIP1 and SOCS1 as the major miR-155 targets. mRNA from BMMs of WT, *miR-155⁻/⁻*, *miR-146a⁻/⁻* and double knock out (DKO) mice before and 8 h after LPS stimulation was subjected to RNA sequencing. **a** Heat map, indicating the number of differentially expressed genes between samples of all strains before (top) and 8 h after LPS stimulation (bottom). **b** Enriched functional annotations for genes upregulated in stimulated *miR-146a⁻/⁻* BMMs compared to WT 8 h after LPS stimulation. **c** qPCR quantification of potential miR-146a and miR-155 targets in BMMs 24 h after LPS stimulation. **d** Protein from BMMs of WT, *miR-155⁻/⁻*, *miR-146a⁻/⁻* and dKO mice was subjected to western blot analysis 24 h after LPS stimulation. **e–h** Mice were reconstituted with bone marrow from WT, *miR-155⁻/⁻*, *miR-146a⁻/⁻* or dKO mice transduced with GFP expressing control (MG), SHIP-1 or SOCS-1 shRNA vectors. **e** Reconstitution competitiveness of SOCS1 and SHIP1 attenuated WT bone marrow cells compared to control as measured by GFP expressing CD45⁺ cells. **f** Ki67 and Hochst proliferative state of CD45⁺ WT cells transduced with SOCS1 shRNA compared to control. **g** CD11b⁺ GFP⁺ peripheral blood macrophage percentage of WT cells transduced with control, SHIP1 shRNA or SOCS1 shRNA. **h** CD11b⁺ peripheral macrophage percentage comparison between WT, *miR-155⁻/⁻*, *miR-146a⁻/⁻* and dKO mice transduced with SHIP1 shRNA vector. (**a**, **b**) $N = 10$, (**c**, **d**) $N = 8$ from three independent experiments, (**e–h**) $N > 7$ per group from two independent experiments. **c**, **e**, **g**, **h** Presented mean ± SEM. *$p < 0.05$, **$p < 0.01$, using one-way ANOVA

mice compared to *miR-155⁻/⁻* and DKO BMMS (Supplementary Fig. 5B and Fig. 4c, d).

Three months after reconstitution, WT mice transduced with *Socs1* shRNA had elevated numbers of CD45⁺ GFP⁺ cells compared to control mice, indicating a slight proliferative advantage to haematopoietic cells with downregulated levels of *Socs1*. Indeed, using proliferation markers, we showed

that CD45⁺, as well as CD11b⁺ cells from WT mice transduced with *Socs1* shRNA are more proliferative than cells from control mice (Fig. 4e, f; Supplementary Fig. 5C). However, *Socs1* downregulation did not lead to a myeloid bias since all cell types proliferated to a similar extent. Downregulation of *Ship1*, on the other hand, led to a slight decrease in engraftment compared to control mice, leading to lower percentage of *Ship1* shRNA cells

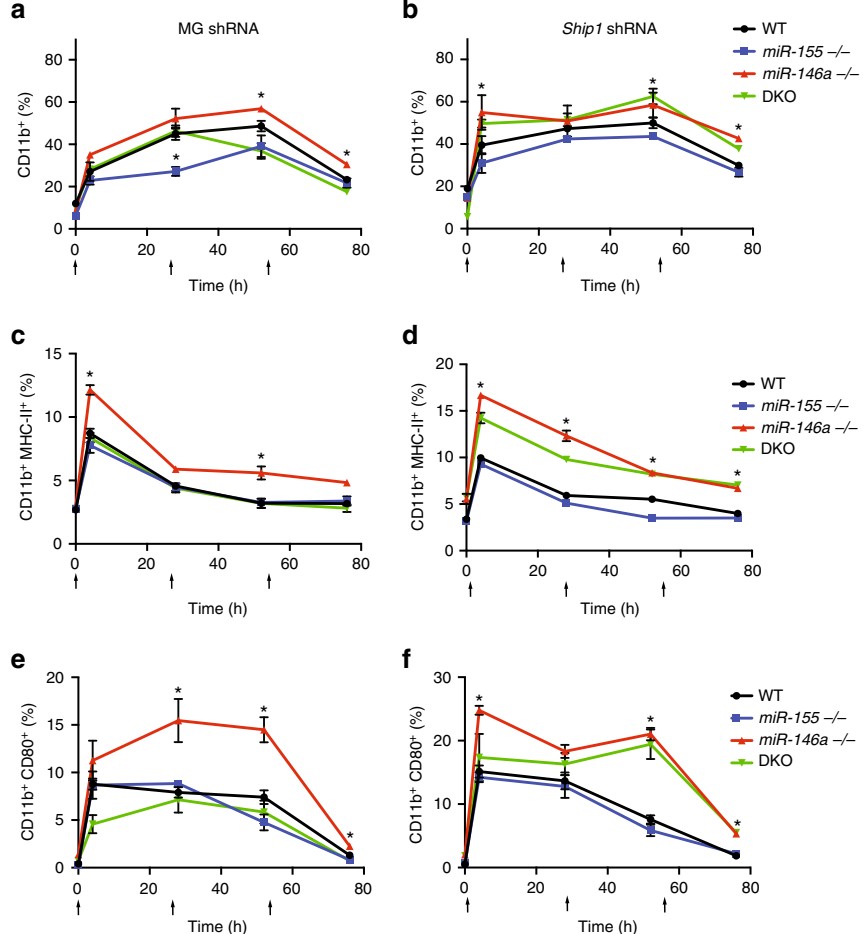

**Fig. 5** SHIP1 downregulation restores *miR-146a*$^{-/-}$ phenotype in DKO macrophages. WT, *miR-155*$^{-/-}$, *miR-146a*$^{-/-}$ and double knock out (DKO) bone marrow cells transduced with control (MG) (left) or SHIP1 shRNA (right) were reconstituted into recipient mice. Three months after reconstitution, the mice received 1 mg/kg LPS every 24 h for 3 days via IP injections. Peripheral blood macrophage CD11b$^{+}$ percentage (**a**, **b**), MHC-II high— (**c**, **d**) and CD80$^{+}$ (**e**, **f**) were compared for all strains by FACS analysis. Arrows indicate time of LPS injections. N > 7 per group from two independent experiments, represented as mean ± SEM. *p < 0.05, using two-way ANOVA

(Fig. 4e). However, *Ship1* downregulation led to a significant myeloid bias in all backgrounds, as shown by an increase in numbers and percentage of CD11b$^{+}$ similar to control *miR-146a*$^{-/-}$ reconstitution (Fig. 4g, h). Importantly, attenuating *Ship1* expression in DKO leads to an even higher myeloid bias compared to *miR-155*$^{-/-}$ or WT donors, which resembled *miR-146a*$^{-/-}$ *Ship1* attenuation (Fig. 4h). These results imply that *Ship1* repression is key to the *miR-146a*$^{-/-}$ myeloid phenotype, while *Socs1* repression is important for the proliferative phenotype.

To determine the roles of *Ship1* and *Socs1* regulation by miR-155 in the acute inflammatory response in vivo, we serially injected 1 mg/kg LPS every 24 h for 3 days and monitored peripheral blood CD11b$^{+}$ macrophage activation 4 h after each injection. Mice reconstituted with the control or *Socs1* shRNA vectors showed similar responses to the miRNA KO mice, where *miR-146a*$^{-/-}$ mice displayed significantly more CD11b$^{+}$ cells as well as higher MHC-II and CD80 activation markers compared to WT control mice (Figs. 1 and 5a, c, e). On the other hand, when mice were reconstituted with the *Ship1* shRNA vector, both *miR-146a*$^{-/-}$ and DKO presented significantly higher activation markers compared to *miR-155*$^{-/-}$ and WT control mice, showing that *Ship1* is a key target of miR-155 during the inflammatory response (Fig. 5b, d, f). SHIP1 is an inositol polyphosphate-5-phosphatase that hydrolyses the 5-phosphate of phosphatidyli-nositol-3,4,5-trisphosphate (PtdIns(3,4,5)P3) to produce PtdIns (3,4)P2[31]. SHIP1 was described as a negative regulator of the inflammatory response by negatively regulating the PI3K-AKT (phosphoinositide 3-kinase) as well as other inflammatory pathways[29,32–34]. PI3K-AKT signalling has been extensively described as both a positive and negative regulator of NF-κB activity depending on the cell type and environmental condi-tions[35–39]. To further understand SHIP1 regulation of PI3K-AKT signalling and NF-κB activity in our system, we measured the protein levels of SHIP1, pAKT, pIKKα/β and phosphorylated p65. We found that *miR-146a*$^{-/-}$ BMMs, which have lower levels of SHIP1, expressed higher levels of pAKT as well as pIKKα/β and phosphorylated p65 compared to WT BMMs. By contrast, *miR-155*$^{-/-}$ and DKO BMMs express higher SHIP1 levels and have a mild reduction in pAKT, pIKKα/β and phosphorylated p65 when stimulated with LPS (Fig. 4d). Of note, TLR4 protein levels remained similar in all strains after LPS stimulation, suggesting that the regulation of NF-κB activity is downstream of the receptor. Together, these results imply that in our experimental setting, both *Socs1* and *Ship1* are primary targets of miR-155. We show that SOCS1 functions as a proliferation repressor in all haematopoietic cells examined, and that SHIP1 functions as a repressor of the inflammatory response and NF-κB activity. *Ship1*

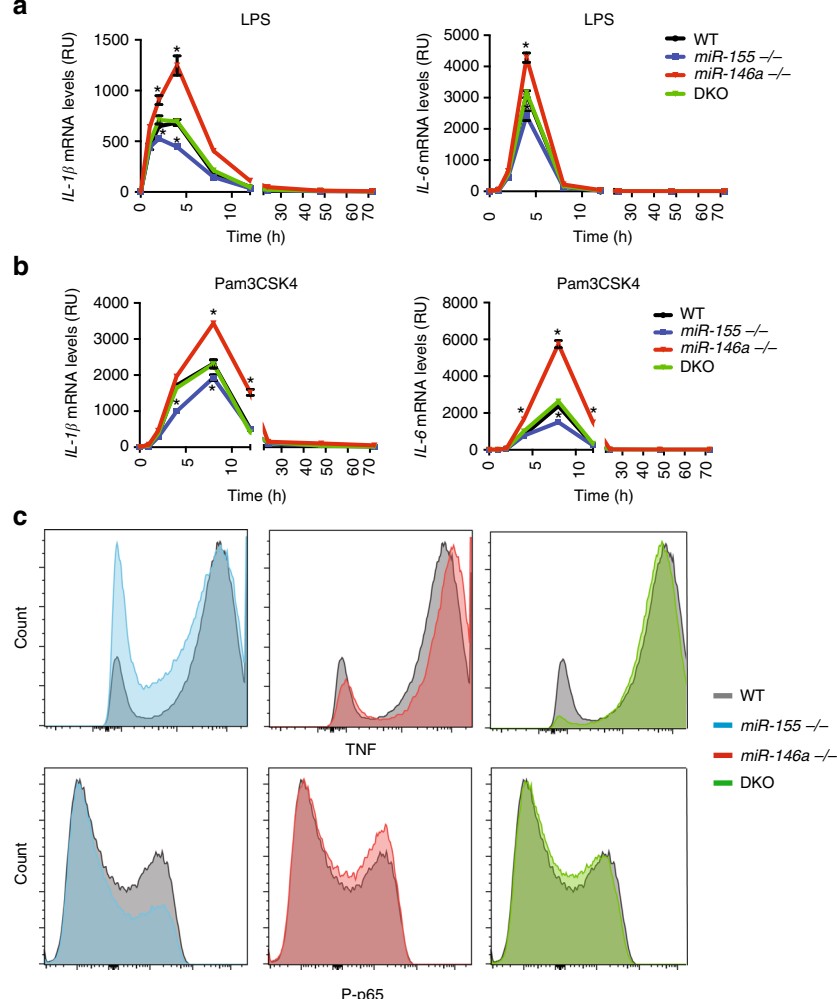

**Fig. 6** Mir-155 acts as an amplifier in suboptimal stimulation. WT, *miR-155*$^{-/-}$, *miR-146a*$^{-/-}$ and double knock out (DKO) BMMs were stimulated with low levels of LPS (10 ng/mL) (**a**) or Pam3CSK4 (5 ng/mL) (**b**). mRNA levels of *IL-1*β (left) and *IL-6* (right) were then quantified using qRT-PCR. **c** TNF and phospho-p65 (P-p65) protein expression of BMMs from WT (grey), *miR-155*$^{-/-}$ (blue), *miR-146a*$^{-/-}$ (red) and dKO (green) 4 h after stimulation with low levels of LPS (10 ng/mL) as measured by FACS using intracellular staining. $N = 4$ per group from at least two independent experiments, represented as mean ± SEM. **a**, **b** *$p < 0.05$, using two-way ANOVA

downregulation by miR-155 enhances NF-κB activity, at least in part, by alleviating the repression on PI3K-AKT signalling. Attenuated PI3K-AKT signalling leads to reduced pIKKα/β phosphorylation and reduced NF-κB activity following LPS stimulation.

**miR-155 and miR-146a combined regulation of NF-κB activity.** We showed that *miR-146a*$^{-/-}$ mice and BMMs display a stronger and longer inflammatory response to LPS, indicating the crucial role of miR-146a in regulating the duration and amplitude of this response. In addition, we observed minor changes in the inflammatory response of miR-155-deficient mice and BMMs when stimulated with LPS (Fig. 2), which indicate that this positive regulator plays a minor role in the strength of the inflammatory response in saturated stimulation conditions. To better understand miR-155 function in the regulation of the macrophage acute inflammatory response, we stimulated BMMs with low levels of LPS (10 ng/mL) or Pam3csk4 (5 ng/mL), which are closer to endogenous levels during sepsis[40]. We then compared expression of TNF, IL-6 and IL-1β using qPCR. We found that with low levels of inflammatory stimuli, miR-155-deficient BMMs express lower levels of inflammatory cytokines compared

to WT controls (Fig. 6a, b). These results were also verified using FACS quantifying intra-cellular levels of TNF and phosphorylated p65 (Fig. 6c). Taken together, these results now show that miR-155 acts as an inflammatory amplifier in suboptimal stimuli, enabling a robust innate response in a wide range of concentrations and stimulations.

We next examined the temporal expression of miR-146a and miR-155 during an inflammatory response using BMMs from WT mice. We found that the dynamics of miR-155 and miR-146a expression correlated with the acute response after exposure to high levels of LPS or Pam3CSK4, whereby miR-155 levels rose rapidly, reaching peak levels at 12 h and gradually decreased from 24 to 48 h post stimulation. miR-146a levels, on the other hand, started to accumulate only after 8 h, reaching peak levels around 24 h and remained highly expressed for up to 72 h post stimulation. In the time frame of 2–24 h post stimulation, we observed most of the transcription of the inflammatory cytokines *IL-1*β and *IL-6*, representing the macrophage inflammatory response. By 40 h, *IL-1*β and *IL-6* levels had fallen to near basal levels (Fig. 7a, b). Similar relative kinetics, though with different time scales, were observed with different kinds of inflammatory stimuli, such as poly I:C and *TNF* (Supplementary Fig. 6A, B). This orchestrated dynamics in the temporal expression of miR-

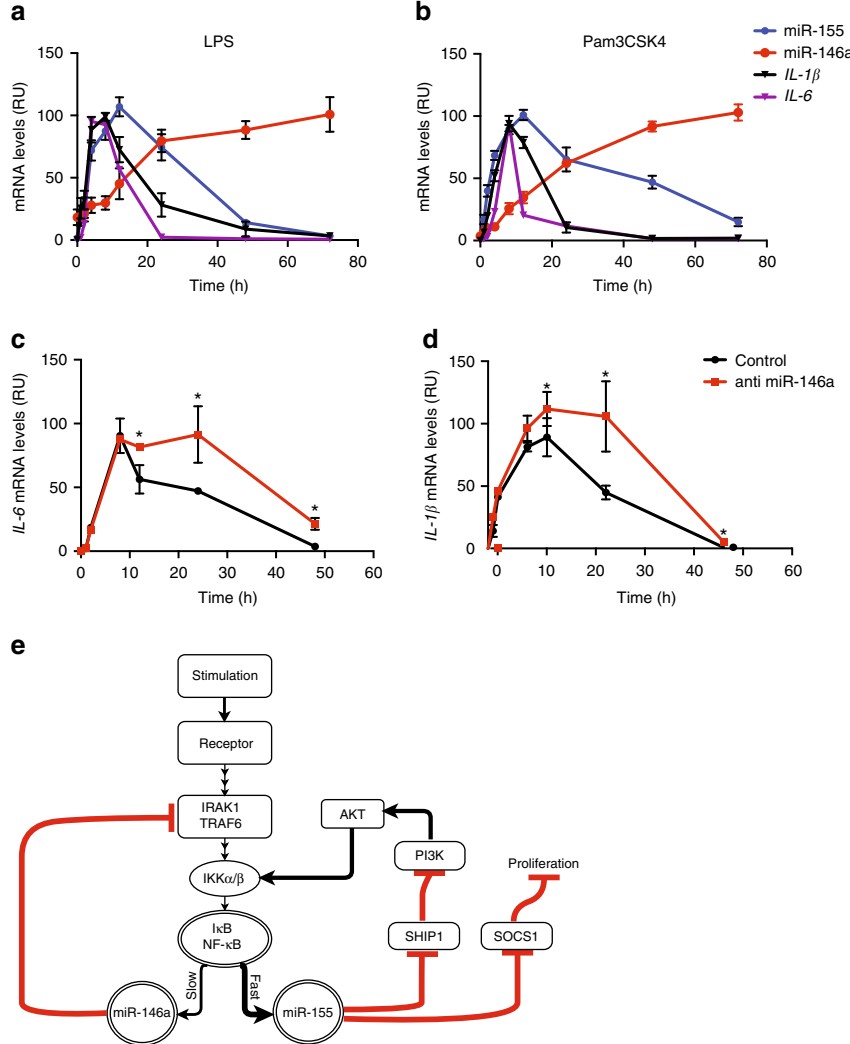

**Fig. 7** MiR-155 and miR-146a form a combined positive and negative auto regulatory loop to control precise NF-κB activity during inflammatory stimuli. The dynamic expression of miR-155, miR-146a, and the NF-κB targets *IL-1β* and *IL-6* at different time points after LPS (**a**) and Pam3CSK4 (**b**) stimulation. The dynamic expression of *IL-6* (**c**) and *IL-1β* (**d**) mRNA in WT BMMs transfected with miR-146a specific anti miR (red) or control (black) 2 h post LPS stimulation (100 ng/mL). **e** A suggested model for the miR-146a and miR-155 regulatory loop during inflammatory stimulation. **a**, **b** $N = 4$ per group from two independent experiments. *$p < 0.05$, using one-way ANOVA (**c**, **d**)

146a and miR-155 with NF-κB activity during TLR activation present a defined time frame for optimal macrophage inflammatory response and its resolution.

To further establish the role of miR-146a in the resolution of the inflammatory response at late stages, miR-146a was knocked down in WT BMMs 2 h post LPS stimulation. miR-146a knockdown BMMs expressed *IL-1β* and *IL-6* mRNA comparably to control for up to 8 h post stimulation. Starting 12 h post stimulation, both *IL-1β* and *IL-6* mRNA levels continued to increase, and remained higher than control for up to 48 h post stimulation, similarly to the behaviour of *miR-146a*$^{-/-}$ BMMs (Fig. 7c, d).

Together, these results indicate that miR-155 and miR-146a form a combined positive and negative regulatory loop controlling NF-κB activity, where inflammatory stimuli activate NF-κB, which rapidly activates miR-155 expression. miR-155 then acts as an amplifier and positive regulator to ensure robust and strong NF-κB activity. With time, miR-146a levels rise to negatively regulate NF-κB activity, leading to attenuation of miR-155 expression and resolution of the inflammatory response (Fig. 7e).

## Discussion

The innate immune response functions as the first line of defense against pathogens. In many cases, pathogens are cleared solely by the innate system without the requirement of the adaptive response. In order to do so, it is crucial that the inflammatory response be efficient and fast, responding optimally to various pathogens in a wide range of concentrations. At the same time, the inflammatory response has to be self-limited to ensure resolution.

In this work, we characterized a miRNA-based regulatory network that enables tight regulation of an 'on' and 'off' switch of NF-κB activity and the macrophage inflammatory response. We show that an inflammatory stimulus leads to the activation of NF-κB, which in turn activates miR-155 transcription. The rapidly and highly transcribed miR-155 represses the expression of SHIP1 and SOCS1 (among other potential targets), amplifying NF-κB activity and enabling a proliferative state for robust and strong macrophage activation. As the inflammatory response develops, miR-146a levels accumulate in a delayed manner leading to the repression of IRAK1 and TRAF6, thus attenuating the signals to NF-κB activation. NF-κB activity is

therefore reduced, leading to abrogation of the transcription of inflammatory genes as well as miR-155. miR-155 levels diminish within 24 h after stimulation, while miR-146a levels remain high and stable for the entire duration of the response. The rapid attenuation of miR-155 levels enables upregulation of SHIP1 and SOCS1 expression, enforcing the repression of NF-κB activity and the inflammatory response, ensuring resolution (Fig. 7e).

Previous investigations using mathematical modelling and bacterial experiments have described a similar regulatory network architecture, where a rapid positive regulator precedes a delayed negative feedback. In these systems, this regulatory network was shown to create a pulse response with a defined 'on' and 'off' characteristic[41,42]. We show that in our case, two miRNAs participate in a combined positive and negative regulatory loop to enable a precise, pulse-like acute inflammatory response, which shortens the time to maximum inflammation levels, as well as the time to resolution.

As the negative regulator, miR-146a contributes to the regulation that determines the amplitude and duration of the response. miR-155 functions as an amplifier, providing a strong initiation of a response even in the face of noise, different kinds of stimuli and various concentrations, as well as a general amplification in suboptimal stimulus conditions.

Our work also emphasizes the importance of the interaction of the two miRNAs because of miR-146a-mediated regulation of miR-155 expression, and the dominant role miR-155 expression has on the development of chronic inflammation. We show that when miR-146a is ablated, miR-155 levels are elevated during the inflammatory response as well as in steady-state conditions (Supplementary Fig. 6C). miR-155, in turn, represses the expression of SHIP1 and SOCS1 and prevents their repression of the inflammatory response. This constant expression of miR-155 contributes, with time, to the low-grade inflammatory status found in miR-146a$^{-/-}$ mice.

It has been shown that miR-146a deficiency can contribute to several human diseases, such as 5q-syndrome, and various types of cancer such as prostate, breast and ovarian[43–46]. Elevated miR-155 expression was also shown to correlate with multiple kinds of cancers, as well as autoimmune and diseases neurologic disorders such as Alzheimer's disease[47–51]. Our work now shows that these two similar phenotypes appear to be different sides of the same coin.

We and others have shown that in miR-146a$^{-/-}$ mice, several cell types contribute to the myeloproliferative and cancer phenotype. Among them Th1, T regulatory and myeloid cells have been described[13,52,53]. We show here that deleting miR-146a in the myeloid lineage alone can recapitulate these phenotypes found in total KO mice. This implies that the myeloid lineage plays an essential role in the initiation and progression of the pathologies found in the total KO mouse model.

It is known that expression of both miR-155 and miR-146a is dependent on NF-κB activity[12,54], but we now show that there is a temporal separation in their expression dynamics, which enables this specific regulatory architecture. We show that during an inflammatory response, although the entire response is mediated by NF-κB, individual induced genes may have very different kinetics of response. The exact mechanism of temporal control for these miRNA genes remains to be fully determined, but could involve combinatorial binding of several transcription factors in addition to NF-κB, as well as different half-life kinetics and stability, chromatin modifications and nuclear architecture around these two miRNAs[55–57]. Further studies are required to better understand the mechanisms regulating this temporal expression that ensures a tight regulation on NF-κB activity in different cell types.

Using a short seed recognition sequence, miRNAs are capable of negatively regulating the expression of several genes simultaneously, enabling regulation of several components within a signalling pathway in the same cell. miRNAs can also regulate different genes and pathways in different cell types, depending on the transcriptome milieu. This regulatory network of NF-κB, miR-155 and miR-146a can therefore be utilized in different cell types and different conditions, potentially resulting in different consequences. Indeed, several targets for both miR-155 and miR-146a have been described, depending on the cell type and condition[23,24,44,51,53,58].

In summary, our data suggest that miR-155 and miR-146a form a unique regulatory network motif to ensure a precise macrophage inflammatory response via regulation of NF-κB activity. It also sheds light on the molecular hierarchy and interaction of these two miRNAs during an inflammatory response, where miR-146a is essential for downregulating miR-155, preventing the deleterious effects of its constant elevated expression.

## Methods

**Mice.** The California Institute of Technology Institutional Animal Care and Use Committee approved all experiments. C57BL/6 WT, *LyzM-Cre miR-146a$^{fl/fl}$*, miR-146a$^{-/-}$, miR-155$^{-/-}$ and DKO as well as NF-κB reporter mice were bred and housed in the Caltech Office of Laboratory Animal Resources facility in specific pathogen-free conditions. Bone marrow reconstitution experiments were performed as described below and in ref. [23] with the mentioned vectors. Recipient mice were monitored for health, and peripheral blood was analysed for mature blood cell types and activation markers each month or after 4 h after each LPS injection up till the experimental end point at either 16 or 36 weeks post reconstitution. At each end point, immune organs were collected for further analysis as described. The number of mice for each experimental cohort and number of experimental repeats are described in the figure legends.

**Listeria monocytogenes infections.** *Listeria monocytogenes* (strain 10,403 serotype 1) were grown in brain heart infusion media. A total load of 10E5 c.f.u. were injected to each mouse using retro-orbital injections. Mice were housed for 3 days, then culled using CO$_2$. Liver, blood and spleen tissues were collected for bacterial load quantification, haematopoietic cell frequencies and IL-6 serum levels.

**S. Typhimurium infection.** Bone marrow cells were cultured for 6 days in BMM media and differentiated to BMMs. On day 6, BMMs were infected with and *S.* Typhimurium at a 10:1 ratio. One hour post *S.* Typhimurium infection, BMMs were washed and media was replaced to BMM media with gentamicin (100 μg/mL) to kill extracellular bacteria. BMMs were collected 24 h post infection and subjected to FACS analysis.

**DNA constructs.** For in vivo miR-155, miR-146a overexpression and SHIP1 as well as SOCS1 shRNA experiments, the mature miR-155 and miR-146a, or SHIP1 and SOCS1 shRNA sequence was synthesized in the miRNA-155 loop-and-arms format[59] and cloned into the MSCV-eGFP (MG) or MSCV-IRES- Th1.1 vectors. dmiR vector was constructed by inserting miR-146a and miR-155 in the pre-miR-17-92 cluster sequence. All miRNAs and shRNA sequences are depicted in Supplementary Table 2.

**Bone marrow reconstitution.** WT C57BL/6 or NF-κB reporter mice were treated with 5-fluorouracil (10 μg; Sigma) for 5 days to enrich for haematopoietic stem and progenitor cells (HSPCs) in the bone marrow. After 5 days, bone marrow cells were collected, red blood cells (RBCs) were lysed with RBC lysis buffer (BioLegend), and cells were plated in HSPC media, which was comprised of complete RPMI with mouse SCF (50 ng/mL), IL-3 (20 ng/mL) and IL-6 (50 ng/mL). Cells were then cultured in 24-well plates for 24 h and spin-infected with PCL-ecotropic pseudo-typed gamma-retrovirus expressing the construct of interest, which was either a miRNA or shRNA, as described in main text. Spin infections were performed by removing supernatant carefully from cell culture plates and adding virus with 8 μg/mL polybrene (Santa Cruz Biotechnology). Plates were then placed in a centrifuge for 2 h at 30 °C and 2500 r.p.m. Immediately following infection, virus supernatant was removed and replaced with HSPC media. Twenty-four hours later, a second identical spin infection was performed. After another 24 h, recipient mice were lethally irradiated (1000 rads from Cs137 source) and 250,000 to a million virus-infected HSPCs were retro-orbitally delivered to reconstitute the immune system. Recipients were maintained on Septra and in autoclaved cages for at least 1 month post reconstitution.

**Cell culture**. Cells were cultured in a sterile incubator that was maintained at 37 °C and 5% $CO_2$. Primary cells were cultured in complete RPMI supplemented with 10% FBS, 100 U/mL penicillin, 100 U/mL streptomycin, 50 μM β-mercaptoethanol and appropriate growth cytokines as needed for the experiment (see below). 293T (ATCC, CRL-3216) cells were cultured in DMEM supplemented with 10% fetal bovine serum, 100 U/mL penicillin and 100 U/mL streptomycin. L929-conditioned media, containing macrophage colony-stimulating factor (M-CSF) essential to BMM maturation, prepared by plating 1.5e6 L929 cells (ATCC, CCL-1) in a 150-cm² flask in 125 mL D10 (DMEM, 10% FBS and 1% Pen/Strep), was grown for 7 days at 37 °C and 5% $CO_2$.

**BMMs culturing**. Mice were culled using $CO_2$. Femur and tibia bones were collected and bone marrow flushed with DMEM. Collected cells were pelleted and resuspended in 10 mL of 1× RBC lysis buffer for 5 min. Cells were resuspended in 20 mL of fresh DMEM. 2e6 bone-marrow cells were plated in a 15-cm tissue culture (TC) dish in 20 mL of BMM media (DMEM, 20% FBS, 30% L929 condition media and 1% Pen/Strep), and grown at 5% $CO_2$ and 37 °C. BMM media was completely replaced on day 3. On day 6, BMMs were replated at a concentration of 1e6 cells per well in a six-well plate and incubated for 16 h. Stimulations and treatments began at day 7.

**miRNA inhibitors transfection**. Bone marrow cells were cultured for 6 days in BMM media and differentiated to BMMs. On day 6, cells were LPS-stimulated (100 ng/mL). Two hours after stimulation, cells were transfected with either miR-146a inhibitor (mirVana miRNA MH10722) or negative control (mirVana miRNA Inhibitor, Negative Control #1) using Lipofectamine RNAiMAX Transfection Reagent (Invitrogen cat: 13778030) according to manufacturer instructions. Cells were then collected at different time points for RNA extraction and quantitative PCR (qPCR) analysis.

**Virus production**. To generate retrovirus for bone marrow cells infection, 10 million HEK293T (ATCC CRL-3216) cells were first plated in a 15 cm plate. Twenty-four hours later, cells were transfected with both the pCL-Eco vector and either the pMG vector or the relevant variant described above for gene delivery. For transfection, we used BioT (Bioland Scientific cat: B01-01) as per the manufacturer's protocol. Thirty-six hours after transfection, virus was collected, filtered through a 45 μM syringe filter and used for infection of HSPCs.

**Expression profiling and qPCR**. We performed real-time qPCR (RT-qPCR) with a 7300 Real-Time PCR machine (Applied Biosystems). Taqman qPCR was performed for miR-155 (002571), miR-146a (478399 miR) and snoRNA-202 (001232 control) detection as per manufacturer's instructions using Taqman MicroRNA Assays (Life Technologies). SYBR Green-based RT-qPCR was performed for mRNA of specific mouse genes. Primers used for qPCR are listed in Supplementary Table 2. The quantification of miR-146a and miR-155 targets following complementary DNA (cDNA) synthesis was done using qScript cDNA SuperMix (Quanta cat: 95048-100) and detection with PerfeCTa qPCR Fastmix with ROX (Quanta cat: 95119-012) as per manufacturer's instructions.

**Sample preparation for RNA sequencing**. BMMs from WT, *miR-155$^{-/-}$*, *miR-146a$^{-/-}$* and DKO were cultured and treated as described above. Cells were lysed using an RNeasy kit (cat no: 74104 Qiagen) with *DNAseI* digestion (cat no: 79254 Qiagen) as per manufacturer's protocol. RNA-seq libraries were prepared from polyA$^+$-selected RNA using the TruSeq RNA Sample Preparation kit (Illumina RS-122-2001) at the Millard and Muriel Jacobs Genetics and Genomics Laboratory at Caltech. Libraries were sequenced on the Illumina HiSeq 2500 generating single-end 50 bp reads. The refSeq annotation for the mm9 version of the mouse genome was used to create a transcriptome Bowtie (version 0.12.7)[60] index. Gene expression levels were estimated using eXpress (version 1.5.0; ref.[61]), and DESeq[62] was used for evaluating differential expression. The raw sequencing reads have been made available under Gene Expression Omnibus (GEO) accession number GSE88791.

**Western blots**. BMM samples were prepared as described for RNA preparation. Cell extracts were collected using RIPA lysis buffer (Sigma cat: R0278-50ML), and were subjected to gel electrophoresis and transfer onto a PVDF membrane. Proteins were detected using the following antibodies: anti-Phospho-Stat3 (#9131 Cell Signaling 1:250), anti-Ship1 (#2728 Cell Signaling 1:200), anti-Phospho-NF-κB p65 (#3033 Cell Signaling 1:300), anti-rabbit IgG HRP (#7074 Cell Signaling 1:10,000), anti-Phospho-AKT (#4060 Cell Signaling 1:250), anti-Phospho-IKKα/IKKβ (#2078 Cell Signaling 1:300), anti-Socs-1 (sc-9021 1:100) and anti-TLR4 (sc-29072 Santa Cruz 1:250). Uncropped images of western blots shown in Fig. 4d are depicted in Supplementary Fig. 8.

**Flow cytometry**. Cells were stained with fluorophore-conjugated antibodies (all from BioLegend unless indicated) in various combinations to characterize relevant haematopoietic cell populations. Intracellular staining was performed by first performing surface staining of cells, followed by fixation and permeabilization

(Cytofix/Cytoperm kit; BD Biosciences) according to manufacturer instructions, and subsequent staining with either Ki67 (BioLegend) and Hoescht33342 (Life Technologies) for cell-cycling analysis, or p-p65 and TNF antibodies (BioLegend) for inflammatory response. Samples were analysed on a MACSQuant10 Flow Cytometry machine (Miltenyi). Gating and analysis was performed using FlowJo software. Gating strategies are depicted in Supplementary Fig. 7.

**Statistical tests**. All statistical analysis was done in Graphpad Prism software using an unpaired Student's t test, one-way or two-way analysis of variance. Data was reported as mean ± SEM. Significant measurements were marked as follows: *$p < 0.05$, **$p < 0.01$, ***$p < 0.001$ or NS for not significant.

**Data availability**. The authors declare that the data supporting the findings of this study are available within the article and its Supplementary Information files, or are available upon reasonable requests to the authors. The raw sequencing data of WT, *miR-146$^{-/-}$*, *miR-155$^{-/-}$* and DKO BMMs before and 8 h after LPS stimulation have been deposited in GEO database under the accession number GSE88791.

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

## Acknowledgements

We thank I. Antoshechkin and V. Kumar at the Caltech Genetics and Genomics Laboratory for their assistance. We thank L.-F. Lu and A. Rudensky for providing *miR-146a fl/fl* mice. This work was supported by an NIH RO1AI079243 (D.B.), the Human Frontiers Science Foundation (M.M.) and National Research Service Award CA183220 (A.M.).

## Author contributions

M.M., J.L.Z. and D.B. designed the study. M.M. conducted all the experimental work with assistance from A.M., J.L.Z., K.L. and Y.G.-F. G.K.M. and M.M. performed bioinformatics analysis. M.M. and D.B. wrote the manuscript with contributions from all authors.

## Additional information

**Competing interests:** The authors declare no competing financial interests.

