## [Peer Review File · Nature Communications]

Reviewers' comments:

Reviewer #1 (LPS, innate programming)(Remarks to the Author):

The authors have characterized the miR-155/miR146a regulatory network that is potentially involved in the modulation of robust and time-limited inflammatory response essential for functional immunity. Overall, the data are well collected and the conclusions are convincing.

Although the involvements of miR-155 and miR-146a during the monocyte/macrophage activation have been recognized in the field and not particularly novel, the combined study of their coupled involvements may have certain novelty.

In terms of the functional readouts for this study, the authors observed altered expression of IL-6, IL-1b, TNF, and altered HSC profiles. It is not clear whether the altered cytokine expressions may be due to altered cellular activation, and/or altered cellular differentiation/apoptosis. The exact functions of miR-155/miR-146a networks in controlling cellular proliferation/differentiation/cell death should be clarified.

The authors conclude that miR-155 may serve as a fast acting network in triggering inflammatory activation, while the miR146a may serve as a time-delayed slow acting network in negatively attenuate the inflammatory activation. However, no experimental data was presented to examine the expression kinetics of these two miRs in vitro and/or in vivo. such data should be presented to further support such claim. Further, miR-146a antagonist should be applied at different time-phases to prove the point that miR-146a is primarily exerting its role at a later phase.

By clarifying these issues, the manuscript may likely provide added novelty to the field of innate immune dynamics.

Additional proof-reading is necessary to clarify the intended studies (e.g. The figure legend for Fig 6 in describing the low levels of LPS mentioned 10 mg/ml. Instead, the methods section of low level LPS mentioned 10 ng/ml).

Reviewer #2 (miRNA, NFkb)(Remarks to the Author):

In this manuscript, the authors report that miR-155 and miR-146a form a combined positive and negative regulatory loop controlling NF- κ B activity and inflammatory responses. They also find that miR-155 expression levels are down-regulated by miR-146a, thus in miR-146a-deficient mice, the elevated miR-155 expression potentiates NF- κ B activity by targeting two inhibitory molecules SHIP1 and SOCS1, which leads to an overactive acute and chronic inflammatory response. The findings are important and interesting, adding insight to systemic understanding of inflammatory response initiation and regulation. However, more experimental data are needed to convincingly support the main conclusions. The concerns need to be addressed as detailed below:

1. In Fig. 2C, serum IL-6 level in miR-155^{-/-} mice challenged with *Listeria monocytogenes* was lower than that in WT mice. Why serum IL-6 level was similar in miR-155^{-/-} mice and WT mice when mice were aged in 12 months (Fig. 1A) or challenged with LPS for 4 h (Fig. 2D).
2. The difference of the inflammatory responses to gram-negative bacteria such as *E. Coli* should also be observed in WT, miR-155^{-/-}, miR-146a^{-/-}, and miR-155^{-/-} miR-146a^{-/-} (DKO) mice.
3. In Fig. 2G, what is the mechanism underlying the up-regulation of miR-155 expression mediated by miR-146 deficiency in LPS-stimulated macrophages? In addition, whether miR-155 deficiency affects

miR-146a expression upon response to innate stimuli?

4. Except for the well-known two molecules SHIP1 and SOCS1, whether does miR-155 target to other novel molecules?

5. In Fig. 4C-D, since miR-146 deficiency up-regulated miR-155 expression, why the mRNA and protein expression of SHIP1 and SOCS1 was not decreased in miR-146a^{-/-} macrophages?

6. Figure 7A-B shows that the dynamic expression of miR-155, miR-146a, and the NF- κ B target IL-1 β at different time points after TLR ligand stimulation. Whether there are similar dynamic expression patterns of miR-155, miR-146a and other inflammatory cytokines such as IL-6 and TNF- α after TLR ligand stimulation, in response to different bacteria challenge or in endotoxin tolerance model.

7. In figure 7A-C, the expression of miR-146a and miR-155 were both induced and regulated by NF- κ B signaling. The expression of miR-155 was increased quickly and maintained for a relative short time, while the expression of miR-146a was increased slowly and maintained for a long time. The authors may discuss the potential mechanism for these differences of expression pattern.

Reviewer #3 (miRNA, myeloid cell function)(Remarks to the Author):

Comments;

[1] novel paper and adds to our understanding of miRNA interplay (generally) and miRNA-146a and miRNA-155 interplay (specifically);

[2] it would be important to add the following recent paper to References and mention in the Discussion:

1: Hill JM, Pogue AI, Lukiw WJ. Pathogenic microRNAs Common to Brain and Retinal Degeneration; Recent Observations in Alzheimer's Disease and Age-Related Macular Degeneration. *Front Neurol.* 2015 Nov 3;6:232. doi: 10.3389/fneur.2015.00232. Review. PubMed PMID: 26579072; PubMed Central PMCID: PMC4630578;

[3]

This recent (2015) paper shows that certain innate immune-system regulatory mRNA 3'-UTRs – (such as the that encoding the human complement factor H; CFH)

(i) that miRNA-146a and miRNA-155 have overlapping binding sites in the CFH mRNA 3'-UTR

and

(ii) that miRNA-146a and miRNA-155 are both involved in human neurological diseases involving inflammatory neurodegeneration [such as Alzheimer's disease (AD) and age-related macular degeneration (AMD)];

.

[4] Suggest publication of this paper after this minor reference addition – and mention in the Discussion.

REVIEWERS' COMMENTS:

Reviewer #1 (Remarks to the Author):

The authors have addressed the concerns properly.

Reviewer #2 (Remarks to the Author):

The authors have addressed the major concerns in the revised manuscript with new data and reasonable revisions.

Reviewer #1 (LPS, innate programming)(Remarks to the Author):

The authors have characterized the miR-155/miR146a regulatory network that is potentially involved in the modulation of robust and time-limited inflammatory response essential for functional immunity. Overall, the data are well collected and the conclusions are convincing.

Although the involvements of miR-155 and miR-146a during the monocyte/macrophage activation have been recognized in the field and not particularly novel, the combined study of their coupled involvements may have certain novelty.

In terms of the functional readouts for this study, the authors observed altered expression of IL-6, IL-1b, TNF, and altered HSC profiles. It is not clear whether the altered cytokine expressions may be due to altered cellular activation, and/or altered cellular differentiation/apoptosis. The exact functions of miR-155/miR-146a networks in controlling cellular proliferation/differentiation/cell death should be clarified.

We thank the reviewer for the remark. During our work, we discovered that the miR-155/miR-146a network controls activation and proliferation of myeloid cells.

In this and in our previous work, we show that aged miR-146a KO mice develop chronic inflammation, and altered HSC profiles and myeloid expansion compared to WT, miR-155 KO and DKO. We also show that this process takes time to develop and becomes apparent only at ~5 months age. Using DKO mice, we now show in Fig 1 that miR-155 deficiency rescues miR-146a KO phenotype showing interaction between these two miRNAs in immune regulation.

In order to understand the initial mechanisms for miR-155/miR-146a network in controlling cellular proliferation/differentiation/ cell death, we did most experiments in young mice before the accumulation of the pathology observed in aged mice.

We show that in these initial stages in young mice, in steady state and after acute inflammatory response (listeria infection and LPS stimulation), cell frequencies for all mature immune cells (macrophages, T cell, B cells etc.) are similar for all strains, without significant differences in cell death (fig S1,2.) and that the major initial cause for the elevated response in miR-146a KO mice is cell activation (figure 2E,F).

To better emphasize the importance of cell activation in early stages, we added proliferation and death measurements for the LPS injection experiment in figure 2D-F, now incorporated into Fig S2. We also included cell proliferation, death and activation for bone marrow derived macrophages challenged with LPS and *Salmonella Typhimurium* infection (Fig S2 S3). These experiments further establish our observation that the initial role for miR-146a/ miR-155 network is to regulate cell activation, and only later there is a myeloid bias and cell proliferation.

We also show that NF-kB activity is changed in miR-155 OE and in the miR-146a, miR-155 Dmir reconstitution experiment indicating that activation is elevated in these strains. We show that with time, miR-155 and dmiR over expression leads to elevated macrophage proliferation (fig 3A) similar to the chronic phenotype observed in miR-146a KO mice. We show that the molecular mechanism responsible for proliferation is at least partially SOCS1 inhibition by miR-155, and that the activation is mediated by SHIP1 inhibition by miR-155 (Fig 4 E-H).

The authors conclude that miR-155 may serve as a fast acting network in triggering inflammatory activation, while the miR146a may serve as a time-delayed slow acting network in negatively attenuate the inflammatory activation. However, no experimental data was presented to examine the expression kinetics of these two miRs in vitro and/or in vivo. such data should be presented to further support such claim.

In Figure 2G and Figure 7 A, B we did present the expression dynamics of miR-155 and miR-146a after stimulation with various TLR agonists in vitro. To better emphasize the expression dynamics we have now added miR-146a expression after LPS stimulation in WT and miR-155 KO BMMs (Fig 2H). We also added the following text: “Both miR-146a and miR-155 are regulated through the activity of the NF- κ B transcription factor ¹¹, and because miR-146a attenuates NF- κ B activity, we wished to determine whether miR-155 expression is affected by miR-146a repression. For that, we quantified miR-155 expression dynamics in BMMs of WT and *miR-146a*^{-/-} mice following a single dose of LPS. As shown in Fig. 2G, in WT BMMs, miR-155 reaches a peak level by 12hrs and returns to near basal levels by 48hrs. In *miR-146a*^{-/-} BMMs, miR-155 rises more rapidly and declines more slowly; miR-155 stays at what would be peak WT levels even after 48hr, a time when WT macrophages have returned to baseline. These results indicate that miR-146a is a major controller of miR-155 expression during the inflammatory response and that elevated and prolonged expression of miR-155 correlates with an elevated inflammatory response and chronic macrophage activation. We next quantified miR-146a expression dynamics in WT and *miR-155*^{-/-} BMMs. As shown in Fig 2H, in WT BMMs, miR-146a levels rise slower, starting at 8hr post stimulation, peak at 24hr and maintain peak levels until 48hrs. In *miR-155*^{-/-} BMMs, miR-146a expression dynamics followed that of WT BMMs with a slight attenuation starting 36hr after stimulation. These results imply that miR-146a functions at later stages following inflammatory stimulation, and that miR-155 expression might play a role in maintaining prolonged, high levels of miR-146a. “

Further, miR-146a antagonist should be applied at different time-phases to prove the point that miR-146a is primarily exerting its role at a later phase.

We thank the reviewer for the suggestion. We did a time course analysis using miR-146a anti-miR and found that transfecting WT BMMs 2hr post LPS stimulation leads to control behavior for up to 8hr post stimulation, then the inflammatory response follow *miR-146a*^{-/-} BMMs trend as measured by IL-6 and IL-1b mRNA levels. We have now incorporated this result into the main text in the following paragraph: “To further establish the role of miR-146a in the resolution of the inflammatory response at late stages, *miR-146a* was knocked down in WT BMMs 2hrs post LPS stimulation. miR-146a knockdown BMMs expressed *IL-1 β* and *IL-6* mRNA comparably to control for up to 8hr post stimulation. Starting 12hr post stimulation, both *IL-1 β* and *IL-6* mRNA levels continued to increase, and remained higher than control for up to 48hr post stimulation, similarly to the behavior of *miR-146a*^{-/-} BMMs (Fig 7C).

By clarifying these issues, the manuscript may likely provide added novelty to the field of innate immune dynamics.

Additional proof-reading is necessary to clarify the intended studies (e.g. The figure legend for Fig 6 in describing the low levels of LPS mentioned 10 mg/ml. Instead, the methods section of low level LPS mentioned 10 ng/ml). This was fixed, thank you for pointing that out.

We thank the reviewer for all the helpful remarks.

Reviewer #2 (miRNA, NFkb)(Remarks to the Author):

In this manuscript, the authors report that miR-155 and miR-146a form a combined positive and negative regulatory loop controlling NF- κ B activity and inflammatory responses. They also find that miR-155 expression levels are down-regulated by miR-146a, thus in miR-146a-deficient mice, the elevated miR-155 expression potentiates NF- κ B activity by targeting two inhibitory molecules SHIP1 and SOCS1, which leads to an overactive acute and chronic inflammatory response. The findings are important and interesting, adding insight to systemic understanding of inflammatory response initiation and regulation. However, more experimental data are needed to convincingly support the main conclusions. The concerns need to be addressed as detailed below:

1. In Fig. 2C, serum IL-6 level in miR-155^{-/-} mice challenged with *Listeria monocytogenes* was lower than that in WT mice. Why serum IL-6 level was similar in miR-155^{-/-} mice and WT mice when mice were aged in 12 months (Fig. 1A) or challenged with LPS for 4 h (Fig. 2D).

Throughout our work, we show that miR-155 deficiency does not significantly change macrophage immune response. This is shown both under chronic, aging state and acute LPS and TLR stimulations. We did find several instances where miR-155 KO did have a slight but significant difference from WT mice, as can be seen in peripheral blood and spleen macrophages in aged mice (Fig 1D,E), IL-6 levels 3 days after *Listeria* infection (Figure 2C), and 30hrs after LPS injections (Figure 2D).

Serum levels in miR-155 aged mice are similar to WT because this is the basal state where miR-146a creates a chronic inflammation phenotype and leads to IL-6 levels above the ELISA detection sensitivity, whereas all WT, miR-155KO and DKO IL-6 levels are below the detection range of the assay (dotted line in Fig 1A). In the case of LPS stimulation, it can be argued that such a high dose overwhelms the macrophage immune response and the effects of miR-155 are not evident. In support of that interpretation, we show that the levels are significantly lower 30hr post LPS injection. We show that at low concentrations, miR-155 KO effects are noticeable, again implying that miR-155 plays a role in promoting the inflammatory response with suboptimal stimulation. As for the *Listeria* infection, the measurement is in the serum 3 days after sub lethal infection that might be low enough to show effects.

2. The difference of the inflammatory responses to gram-negative bacteria such as *E. Coli* should also be observed in WT, miR-155^{-/-}, miR-146a^{-/-}, and miR-155^{-/-} miR-146a^{-/-} (DKO) mice.

We thank the reviewer for the remark. We show the differences in response to LPS and *Listeria* infection in-vivo, and to different TLR ligands (LPS, Poly-I:C, Pam3CSK4), and TNF α in vitro using BMMs from all strains. To show the response to live, gram-negative bacteria, we now included an experiment of BMM infection with *Salmonella Typhimurium* (shown in Fig S2 C-F) and added the following text: "Similar results were also obtained using bone marrow derived macrophages (BMMs) infected with *Salmonella Typhimurium*. Infection with live gram-negative bacteria led to an elevated inflammatory response in miR-146a $^{-/-}$ BMMs, manifested by elevated CD80 and MHC-II cell surface expression compared to WT, miR-155 $^{-/-}$, and DKO BMMs. No significant difference in cell death or proliferation was observed between strains (Fig. S2C-F)."

3. In Fig. 2G, what is the mechanism underlying the up-regulation of miR-155 expression mediated by miR-146 deficiency in LPS-stimulated macrophages? We believe that over activation of NF- κ B by the lack of repression of miR-146a is the mechanism underlying miR-155 over activation and lack of down regulation. This is in fact one of the major findings from this work that lead us to understand the autoregulatory loop on NF- κ B activity. We explain this point in the discussion- "We show that an inflammatory stimulus leads to the activation of NF- κ B, which in turn activates miR-155 transcription. The rapidly and highly transcribed miR-155 represses the expression of SHIP1 and SOCS1 (among other potential targets), amplifying NF- κ B activity and enabling a proliferative state for robust and strong macrophage activation. As the inflammatory response develops, miR-146a levels accumulate in a delayed manner leading to the repression of IRAK1 and TRAF6, thus attenuating the signals to NF- κ B activation. NF- κ B activity is therefore reduced, leading to abrogation of the transcription of inflammatory genes as well as miR-155. miR-155 levels diminish within 24 hours after stimulation, while miR-146a levels remain high and stable for the entire duration of the response. The rapid attenuation of miR-155 levels enables upregulation of SHIP1 and SOCS1 expression, enforcing the repression of NF- κ B activity and the inflammatory response, ensuring resolution (Fig. 7C)."

In addition, whether miR-155 deficiency affects miR-146a expression upon response to innate stimuli? We added a figure (now Fig 2H) showing the expression dynamics of miR-146a in WT and miR-155 KO BMMs. We show that miR-155 deficiency does not dramatically change miR-146a expression, but leads to a slight attenuation in miR-146a expression at 36-48hrs after stimulation. We added the following text: "We next quantified miR-146a expression dynamics in WT and miR-155 $^{-/-}$ BMMs. As shown in Fig 2H, in WT BMMs, miR-146a levels rise slower, starting at 8hr post stimulation, peak at 24hr and maintains peak levels until 48hrs. In miR-155 $^{-/-}$ BMMs, miR-146a expression dynamics followed that of WT BMMs with a slight attenuation starting 36hr after stimulation. These results imply that miR-146a functions at later stages following inflammatory stimulation, and that miR-155 expression might play a role in maintaining prolonged, high levels of miR-146a."

4. Except for the well-known two molecules SHIP1 and SOCS1, whether does miR-155 target to other novel molecules? We did RNA seq as well as qPCR for several potential targets. We show in figure 4C that we verified that only SHIP1, SOCS1 and Bach1 had

significant mRNA expression differences, while PU.1 only showed a trend and Ets1 did not. Previous experiments in our lab have shown that Bach1 does not contribute to the miR-155 phenotype (unpublished). That is the reason we only concentrated on SHIP1 and SOCS1. A more comprehensive list of differentially expressed miR-155 and miR-146a targets that change in our setting can be found in Sup. Table 1b.

5. In Fig. 4C-D, since miR-146 deficiency up-regulated miR-155 expression, why the mRNA and protein expression of SHIP1 and SOCS1 was not decreased in miR-146a^{-/-} macrophages? We thank the reviewer for pointing that out. There is in fact a reduced expression of SHIP1 and SOCS1 in miR-146a KO BMMs: to emphasize that, we added an average quantification indication for all proteins assayed, compared to WT BMMs, in Figure S5A. We also added the following text: “Interestingly, SHIP-1 and SOCS-1 protein levels were slightly but significantly lower in miR-146a^{-/-} BMMs compared to WT, in line with the higher expression of miR-155 in these cells (Fig. S5A, Fig. 2G).”

6. Figure 7A-B shows that the dynamic expression of miR-155, miR-146a, and the NF-κB target IL-1β at different time points after TLR ligand stimulation. Whether there are similar dynamic expression patterns of miR-155, miR-146a and other inflammatory cytokines such as IL-6 and TNF-α after TLR ligand stimulation, in response to different bacteria challenge or in endotoxin tolerance model. We have now included IL-1b, and IL-6 expression after different TLR agonists- LPS, Pam3CSK4 Poly I:C and TNFα. These results are now incorporated into Fig 7 and Fig S6 A and B. We added the following text to the main section: “In the time frame of 2-24 hours post stimulation, we observed most of the transcription of the inflammatory cytokines *IL-1β* and *IL-6*, representing the macrophage inflammatory response. By 40 hours, *IL-1β* and *IL-6* levels had fallen to near basal levels (Fig. 7 A, B). Similar relative kinetics, though with different time scales, were observed with different kinds of inflammatory stimuli, such as Poly I:C, and TNFα (Fig. S6 A,B). This orchestrated dynamics in the temporal expression of miR-146a and miR-155 with NF-κB activity during TLR activation present a defined time frame for optimal macrophage inflammatory response and its resolution.”

7. In figure 7A-C, the expression of miR-146a and miR-155 were both induced and regulated by NF-κB signaling. The expression of miR-155 was increased quickly and maintained for a relative short time, while the expression of miR-146a was increased slowly and maintained for a long time. The authors may discuss the potential mechanism for these differences of expression pattern. We discuss this in the discussion section: “We show that during an inflammatory response, although the entire response is mediated by NF-κB, individual induced genes may have very different kinetics of response. The exact mechanism of temporal control for these microRNA genes remains to be fully determined but it could involve combinatorial binding of several transcription factors in addition to NF-κB, as well as different half life kinetics and stability, chromatin modifications and nuclear architecture around these two miRNAs^{53, 54, 55}. Further studies are required to better understand the mechanisms regulating this temporal expression that ensures a tight regulation on NF-κB activity in different cell types.”

Reviewer #3 (miRNA, myeloid cell function)(Remarks to the Author):

Comments;

[1] novel paper and adds to our understanding of miRNA interplay (generally) and miRNA-146a and miRNA-155 interplay (specifically);

[2] it would be important to add the following recent paper to References and mention in the Discussion:

1: Hill JM, Pogue AI, Lukiw WJ. Pathogenic microRNAs Common to Brain and Retinal Degeneration; Recent Observations in Alzheimer's Disease and Age-Related Macular Degeneration. *Front Neurol.* 2015 Nov 3;6:232. doi: 10.3389/fneur.2015.00232. Review. PubMed PMID: 26579072; PubMed Central PMCID: PMC4630578;

[3]

This recent (2015) paper shows that certain innate immune-system regulatory mRNA 3'-UTRs – (such as the that encoding the human complement factor H; CFH)

(i) that miRNA-146a and miRNA-155 have overlapping binding sites in the CFH mRNA 3'-UTR

and

(ii) that miRNA-146a and miRNA-155 are both involved in human neurological diseases involving inflammatory neurodegeneration [such as Alzheimer's disease (AD) and age-related macular degeneration (AMD)];

[4] Suggest publication of this paper after this minor reference addition – and mention in the Discussion.

We thank the reviewer for the helpful reference. We added it into our discussion with the addition of the following text: “It has been shown that *miR-146a* deficiency can contribute to several human diseases, such as 5q-syndrome, and various types of cancer such as prostate, breast and ovarian^{42, 43, 44, 45}. Elevated miR-155 expression was also shown to correlate with multiple kinds of cancers, as well as autoimmune and diseases neurologic disorders such as Alzheimer's disease^{46, 47, 48, 49, 50}. Our work now shows that these two similar phenotypes appear to be different sides of the same coin.”

“Using a short seed recognition sequence, miRNAs are capable of negatively regulating the expression of several genes simultaneously, enabling regulation of several components within a signaling pathway in the same cell. miRNAs can also regulate different genes and pathways in different cell types, depending on the transcriptome milieu. This regulatory network of NF-κB, miR-155 and miR-146a can therefore be utilized in different cell types and different conditions, potentially resulting in different consequences. Indeed, several

targets for both miR-155 and miR-146a have been described, depending on the cell type and condition^{22, 23, 43, 50, 52, 57}.

REVIEWERS' COMMENTS:

Reviewer #1 (Remarks to the Author):

The authors have addressed the concerns properly.

We thank the reviewer for all comments; we believe it significantly contributed to our manuscript.

Reviewer #2 (Remarks to the Author):

The authors have addressed the major concerns in the revised manuscript with new data and reasonable revisions.

We thank the reviewer for all comments; we believe it significantly contributed to our manuscript.